# TASK-SPECIFIC META-FEATURE SELECTION FOR FEW-SHOT SEGMENTATION

## ABSTRACT

Few-shot segmentation (FSS) aims to segment new category images given only a few labeled samples. Most previous works concentrate on the design of intricate query decoders to perform feature matching or aggregation between the support and query. In this paper, we revisit a widely overlooked aspect of existing FSS methods, *i.e.*, the exploration of fixed pre-trained backbone features. We find that treating all feature channels equally is suboptimal and propose a Task-specific Channel-wise Modulation Network (TCMNet) to focus more attention on task-aware channels, facilitating more effective utilization of pre-trained features. The proposed TCMNet enjoys several merits. First, we design a self-modulation block that injects the gradient information into channel-wise attention layers, thereby enhancing the discriminability between target and background features. Second, a cross-calibration block is introduced to align the support features toward the query according to the target gradient and representations, which mitigates the impact of intra-class diversity. Extensive experimental results on COCO-$20^i$ and Pascal-$5^i$ benchmarks demonstrate that the TCMNet, as a general plugin, consistently achieves significant improvements over different query decoders and also achieves state-of-the-art results. In addition, the decent performance achieved by exploring the backbone features may inspire another direction for developing more comprehensive FSS models.

## 1 INTRODUCTION

Semantic segmentation has achieved conspicuous achievements benefiting from large-scale annotated datasets (Lin et al., 2014; Mottaghi et al., 2014; Kirillov et al., 2023) and elaborate deep-learning techniques (Long et al., 2015; Vaswani et al., 2017; Ronneberger et al., 2015; He et al., 2016). However, the dependence on extensive annotated data constrains the capabilities of segmentation models to predefined training categories, severely limiting their practical applications. To overcome such inherent category sensitivity and in pursuit of human-like intelligence of learning from scarce samples, few-shot segmentation (FSS) (Shaban et al., 2017) is proposed to derive segmentation models capable of quickly generalizing to novel classes.

Concretely, FSS aims at segmenting new category images (*i.e.*, query images) with only a handful of labeled reference images (*i.e.*, support images). Tackling diverse query images with extremely limited support reference poses great challenges as: (1) **Significant intra-class diversity** between support and query targets is frequently encountered, as shown by the two persons in Figure 1. (2) **Cluttered query backgrounds** often contain distractors, such as training classes (Lang et al., 2022a) or similar interfering objects (*e.g.*, colored boxes in Figure 1 (b)). These factors elevate the risk of errors or incompleteness in target segmentation, constituting two fundamental challenges in FSS.

The current top-performing FSS frameworks usually comprise a ImageNet (Russakovsky et al., 2015) pre-trained Siamese backbone (Liu et al., 2020a) to encode support and query images into the shared feature space, as well as a support-guided query decoder to excavate the query target through cross-image feature matching (Shi et al., 2022a; Li et al., 2021; Zhang et al., 2021c) or aggregation (Min et al., 2021; Hong et al., 2022). To alleviate the challenges discussed above, most recent research has delved into the design of the decoder, yielding considerable progress such as prototypical learning-based (Liu et al., 2020b; Li et al., 2021; Wu et al., 2021; Wang et al., 2024) or affinity learning-based decoders (Zhang et al., 2021c; Wang et al., 2023b; Shi et al., 2022a;

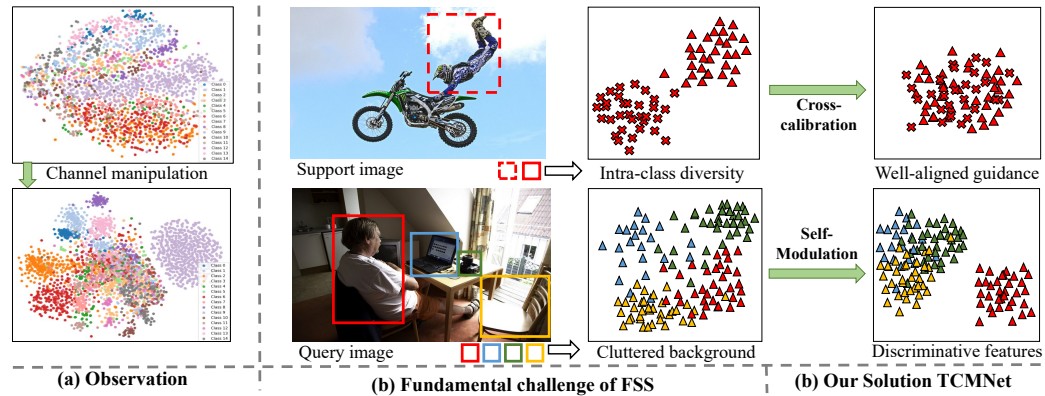

**(a) Observation** | **(b) Fundamental challenge of FSS** | **(b) Our Solution TCMNet**

Figure 1: (a)T-SNE visualization of foreground prototypes of all the training samples. We find that channel manipulation can enhance the discriminability of specific categories. (b)Illustration of two fundamental challenges of the FSS task. (c) Illustration of the feature modulation process.

Min et al., 2021; Peng et al., 2023). Meanwhile, the pre-trained Siamese backbone is typically frozen during the training and testing processes to prevent model overfitting on small datasets, thus facilitating generalization on widely distributed categories. *However*, through an in-depth analysis of backbone features pre-trained via classification objectives, we argue that **employing the pre-trained backbone features straightforwardly is suboptimal**. In fact, multiple channels of backbone features respectively model distinct levels of meta-characteristics. As illustrated in Figure 2(a), the fully supervised classification or segmentation models (Long et al., 2015; Ronneberger et al., 2015) are equipped with category-customized classifiers (fully connected layer or convolutional head) to adaptively combine different channels with various weights for discriminative prediction. While in current FSS approaches, all channels of input features are of equal importance when they are fed into the FSS query decoders (as shown in Figure 2 (b)). This exacerbates the principal challenges of FSS because the meta-characteristics shared across classes might be interfering factors in distinguishing query targets from cluttered backgrounds, and the intra-class meta-characteristics discrepancy implies the potential bias in support guidance. Therefore, exploring a more reasonable utilization of backbone features may offer another avenue for effective FSS.

Drawing upon insights from the realm of feature visualization (Zhou et al., 2016; Selvaraju et al., 2016), we deem that focusing on specific feature channels can effectively enhance the discriminability of features to corresponding categories. As illustrated in Figure 1(a), after randomly dropping some channels of backbone features, there emerges a category that exhibits notable distinguishability from others. Such explicit feature adjustment can serve as an ideal solution tailored for FSS-like binary segmentation. Nevertheless, in the absence of category-customized classifiers in the FSS scenario, a natural question arises: *How to identify and focus on category-related channels when tackling objects of a specific class?*

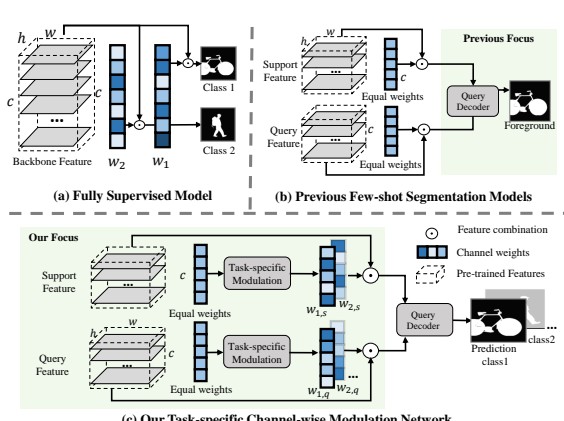

Figure 2: Comparison of how backbone features are used in fully supervised segmentation models (a), previous FSS models (b), and our TCMNet (c).

Driven by this question, in this work, we carefully design the **T**ask-specific **C**hannel **M**odulation (**TCM**) network, which can be applied as a generic plugin to adaptively highlight category-relevant parts of the backbone features before processing by the query decoder as shown in Figure 2(c). TCM coherently alleviates the impacts of the two foundational FSS challenges by incorporating a Self-Modulation Block (SMB) and a Cross-Calibration Block (CCB). Specifically, **to deal with cluttered background**, inspired by deep

explanation methods (Guidotti et al., 2018), we resort to the gradient information to evaluate the importance of different channels for the current task, which is then injected into the channel-wise attention layer as explicit guidance, facilitating the concentration on the task-relevant channels. **To deal with intra-class diversity**, in CCB, we introduce a dual calibration strategy that incorporates two support-to-query channel transformation matrices to adjust the support features to align with the query features. The matrices are respectively built upon the gradient vectors and holistic target representations, serving as the bridge of the intra-class feature gap. Through the synergy of SMB and CBB, the proposed TCMNet not only enhances the target awareness of support and query backbone features but also reconciles the inherent tension between them. Superior feature matching or aggregation within the query decoder can then be achieved on the basis of optimized backbone features.

We evaluate the proposed TCMNet on two widely used benchmarks, *i.e.*, COCO-$20^i$ (Lin et al., 2014) and Pascal-$5^i$ (Everingham et al., 2010) with different backbones. Extensive experiments demonstrate that the lightweight TCMNet consistently boosts performance when integrated with various existing FSS query decoders. Furthermore, the explicit modulation also expedites model convergence. **In summary**, our contributions can be concluded as follows: (i) We jump out of the design of query decoder and steer toward a new perspective of FSS, *i.e.*, employing gradient information to modulate the pre-trained backbone features for more reasonable utilization. (ii) We put forward a novel Task-specific Channel Modulation network (TCMNet), that can be integrated into various FSS methods as a general plugin, to coherently tackle two foundational FSS challenges. (iii) Extensive experimental under different settings demonstrate that our TCMNet consistently elevates the performance of several FSS methods and achieves state-of-the-art results.

## 2 RELATED WORK

### 2.1 SEMANTIC SEGMENTATION

Semantic segmentation aims to classify each pixel within the given image into a specific category and has been widely applied to autonomous driving (Kerner, 2016), medical image processing (Ronneberger et al., 2015), and so on. The seminal Fully-Connected Network (FCN) (Long et al., 2015) achieved significant advances in semantic segmentation and inspired a lot of works (Zhao et al., 2017; Xiao et al., 2018; Ronneberger et al., 2015). Numerous architectures enhance context recognition by expanding the receptive field of CNNs through dilated convolutions (Chen et al., 2017; 2018), global pooling (Liu et al., 2015), and pyramid pooling (Chen et al., 2017; Yang et al., 2018). Besides CNN-based architectures, the emergence of the Vision Transformer (ViT)(Dosovitskiy et al., 2020) has spurred the development of transformer-based segmentation models(Strudel et al., 2021; Zheng et al., 2021; Zhang et al., 2022b; 2021d). Notably, MaskFormer (Cheng et al., 2021b) utilizes the transformer decoder (Carion et al., 2020) for mask classification using a set prediction approach. This framework has been refined by numerous subsequent studies (Cheng et al., 2021a; Zhang et al., 2023; Luo et al., 2023; Sun et al., 2023). Among them, the Segment Anything Model (SAM) (Kirillov et al., 2023) proposes the prompt segmentation paradigm and achieves astonishing segmentation performance after training on extremely huge datasets. Despite their success, these methods struggle to generalize to novel classes in low-data scenarios.

### 2.2 FEW-SHOT SEMANTIC SEGMENTATION

Few-shot segmentation (FSS) (Shaban et al., 2017) is designed to segment new category images with only a few labeled samples as references. Most of the recent FSS frameworks consist of two fundamental components, *i.e.*, a Siamese backbone (Liu et al., 2020a) to extract features and a query decoder to excavate the target within the query image under the guidance of support features. Current researches mainly focus on the design of the query decoder, which can be roughly divided into two categories: prototypical learning decoders and affinity learning decoders. Inspired by PrototypicalNet (Snell et al., 2017), prototypical learning decoders adopt a single (Zhang et al., 2020; Wang et al., 2019; Cao et al., 2022; Liu et al., 2022c; Jiao et al., 2022) or multiple prototypes (Lang et al., 2022b; Yang et al., 2020; Liu et al., 2022b;a; Zhang et al., 2021a; 2022a; Okazawa, 2022; Wang et al., 2022) to represent the target and then conduct feature comparison or aggregation to mine the query target. To capture fine-grained support information, affinity learning decoders (Wang et al., 2020; Min et al., 2021; Hong et al., 2022; Wang et al., 2023b) constructs pixel-level associations

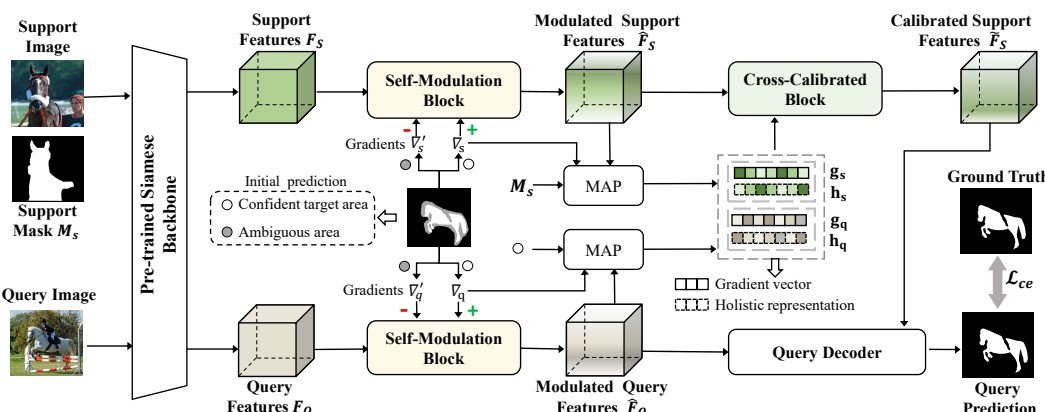

Figure 3: Illustration of the proposed TCMNet. We resort to the gradient information from the initial prediction to local the task-related channels. The gradients are then adopted to guide the attention process within the self-modulation block. The cross-calibrated block employs the gradient vectors and the holistic target representations to align the support features with the query. The processed features possess higher task awareness and lower intra-class diversity, facilitating more reliable feature matching or aggregation within the query decoder.

between query and support features via cost volume aggregation (Min et al., 2021; Hong et al., 2022) or attention techniques (Zhang et al., 2021c; Shi et al., 2022a; Peng et al., 2023). Though achieving promising results, most of these methods only concentrated on the design of the query decoders, neglecting the exploration of backbone features. Some recent works try to solve this problem by fine-tuning (Sun et al., 2022) the backbone or adopting trainable ViTs (Hu et al., 2022). Despite achieving promising performance, it comes with significant additional computational overhead. In this paper, we focus on exploring a more rational utilization of backbone features and introduce a lightweight task-specific channel-wise modulation network (TCMNet) to address the fundamental challenges of FSS from a novel perspective.

## 3 METHOD

### 3.1 PROBLEM DEFINITION

Few-shot Segmentation (FSS) aims to perform novel category object segmentation with only a few densely-annotated samples. Most existing FSS methodologies leverage the meta-training paradigm to enhance the model generalization. Specifically, the datasets are divided into the training set $\mathcal{D}_{train}$ and testing set $\mathcal{D}_{test}$ with class set $\mathcal{C}_{train}$ and $\mathcal{C}_{test}$ respectively. Note that the two class sets are disjoint, *i.e.*, $\mathcal{C}_{train} \cap \mathcal{C}_{test} = \emptyset$. To train the FSS model in the *K*-shot setting (*K*=1 or *K*=5 in this paper), a set of episodes are sampled from $\mathcal{D}_{train}$, each of which consists of the support set $\mathcal{S} = \{I_s^k, M_s^k\}_{k=1}^{K}$ and the query set $\mathcal{Q} = \{I_q, M_q\}$, where $I$ and $M$ denote the RGB image and corresponding ground-truth mask, respectively. The FSS model is optimized to predict the target mask of the query image $I_q$ under the supervision of $M_q$. For testing, the trained model is evaluated on $\mathcal{D}_{test}$ across all the sampled episodes without further optimization.

### 3.2 TASK-SPECIFIC CHANNEL MODULATION NETWORK

#### 3.2.1 OVERVIEW

Most of the previous FSS methods adopt the ImageNet pre-trained backbone to extract the features, keeping it fixed during both meta-training and meta-testing. The extracted features ($F_s$ and $F_Q$ in Figure 3) are fed directly into the query decoder for prediction. We propose modulating the features using gradient information before inputting them into the query decoder. The modulated features ($\hat{F}_s$ and $\hat{F}_Q$ in Figure 3) are less susceptible to the impact of intra-class diversity and cluttered backgrounds. The proposed Task-specific Channel Modulation network (TCMNet), as shown in Figure 3, comprises three major procedures, *i.e.*, 1) channel-wise importance assessment, 2) importance-guided self-modulation, 3) support-to-query cross-calibration. We employ the gradient

information to assess the channel-wise importance of backbone features in procedure 1). In procedure 2), the Self-Modulation Block (SMB) leverages the importance as auxiliary cues of channel-wise attention to enhance the task perceptibility of features. Then, the Cross-Calibration Block (CCB) in procedure 3) adapts the support features to bridge the intra-class feature gap according to the holistic representation as well as grad vector discrepancy. The details are as follows.

### 3.2.2 CHANNEL-WISE IMPORTANCE ASSESSMENT

Meta-characteristics encoded by different feature channels hold differing importance when representing distinct categories. To identify the task-related channels in the absence of category-customized classifiers, motivated by visual explanation techniques (Selvaraju et al., 2016; Guidotti et al., 2018), we employ the gradients to evaluate the importance of different channels for backbone features. Specifically, for a class-agnostic task, we calculate the gradient of the foreground prediction scores for query target pixels, with respect to the support features $\mathbf{F}_s \in \mathbb{R}^{h \times w \times c}$ and query features $\mathbf{F}_q \in \mathbb{R}^{h \times w \times c}$ immediately before the decoder, respectively. Here we denote the corner mark as $*$ and $* \in \{s, q\}$ for concise:

$$\nabla_* = \frac{\partial \mathbf{S}_{fg}}{\partial \mathbf{F}_*} \in \mathbb{R}^{h \times w \times c}, \tag{1}$$

where $\mathbf{S}_{fg}$ denotes the average foreground prediction scores within the query target area, which is calculated based on the two-channel prediction logits through equation 2. It should be noted that we adopt the logits instead of the ground truth mask to determine the foreground area, which alleviates the discrepancy between training and testing, as the query mask is not available at test time, formally,

$$\mathbf{S}_{fg} = \frac{\sum_{(i,j)} \mathbf{P}_{fg}(i,j) \cdot \mathbb{M}(i,j)}{\sum_{(i,j)} \mathbb{M}(i,j))}, \quad \mathbb{M}(i,j) = \begin{cases} 1 & \text{if } \mathbf{P}_{fg}(i,j) - \mathbf{P}_{bg}(i,j) > \delta \\ 0 & \text{otherwise} \end{cases}, \tag{2}$$

the $\mathbf{P}_{fg}$ and $\mathbf{P}_{bg}$ above represent the probabilities of the corresponding query pixels being predicted as foreground and background, respectively. Due to the presence of ambiguous regions in the prediction, we use $\delta$ to select the more confident parts. Additionally, we compute the gradient of these ambiguous regions $\nabla'_*$ according to equation 1 and equation 2, but modify the condition of the $\mathbb{M}(i,j) = 1$ to $|P_{fg}(i,j) - P_{bg}(i,j)| < \delta$. After obtaining $\nabla_*$ and $\nabla'_*$, we employ $ReLU$ to activate the gradients and then adopt spatial average pooling to get the channel-wise weights:

$$\mathbf{G}_* = \text{Average Pool}(ReLU(\nabla_*)), \quad \mathbf{G}'_* = \text{Average Pool}(ReLU(\nabla'_*)). \tag{3}$$

Larger values in $\mathbf{G}_* \in \mathbb{R}^c$ suggest the corresponding channels contribute more to the determination of the current target. Conversely, greater values in $\mathbf{G}'_* \in \mathbb{R}^c$ indicate the channels that may lead to confusion, which are more likely to encode classes-shared meta-characteristics.

### 3.2.3 IMPORTANCE-GUIDED SELF-MODULATION

The self-modulation block (SMB) is designed to enhance the task-relevant channels of backbone features under the guidance of channel-wise importance. It is non-trivial to modulate the backbone features as inappropriate manipulations (*e.g.*, weighting using importance directly) may damage the inherent rich semantic cues as demonstrated in Table 9. In SMB, the $\mathbf{G}_*$ and $\mathbf{G}'_*$ are injected into successive channel-wise self-attention layers to facilitate adaptive feature highlighting. Specifically,

$$\mathbf{A} = \text{Softmax}(\frac{\mathbf{Q}(\mathbf{K})^\mathsf{T}}{\sqrt{\text{d}}} + \lambda(\mathbf{G}_* - \mathbf{G}'_*)), \tag{4}$$

among which the $\lambda$ is a hyper-parameter that controls the proportion of attention and gradient information. The $\mathbf{Q}$ and $\mathbf{K}$ are obtained by:

$$\mathbf{Q} = \varphi(\mathbf{F}_*)\mathbf{W}^\mathcal{Q}, \quad \mathbf{K} = \varphi(\mathbf{F}_*)\mathbf{W}^\mathcal{K}, \tag{5}$$

where $\varphi : \mathbb{R}^{h \times w \times c} \to \mathbb{R}^{c \times hw}$ refers to the reshape function, $\mathbf{W}^\mathcal{Q}$ and $\mathbf{W}^\mathcal{K} \in \mathbb{R}^{hw \times d}$ are learnable projections and $\sqrt{d}$ is the scaling factor. Note that the $\mathbf{G}_*$ and $\mathbf{G}'_*$ are min-max normalized and expanded to the appropriate dimensions before being added to the original attention matrix. The enhanced features are obtained according to the adjusted attention matrix $\mathbf{A}$, and a feed-forward network (**FFN**) is applied to transform the fused features further:

$$\widehat{\mathbf{F}}_* = \varphi^{-1}(\mathbf{FFN}(\mathbf{AV})), \quad \mathbf{V} = \varphi(\mathbf{F}_*)\mathbf{W}^\mathcal{V}, \tag{6}$$

where the $\mathbf{W}^{\mathcal{V}} \in \mathbb{R}^{hw \times d}$ is linear projection. Intuitively, SMB models the channel-wise dependencies with the heightened focus on task-specific channels, tailoring the backbone features for distinguishing targets from the cluttered background.

### 3.2.4 Support-to-query Cross-calibration

To further mitigate bias in support guidance stemming from intra-class feature variations like appearance, scale, pose, etc., the cross-calibration block (CCB) combines a dual strategy to adapt modulated support features $\widehat{\mathbf{F}}_s$ to align with query features $\widehat{\mathbf{F}}_q$. The channel-wise calibration is implemented with two cross-instance transformation matrices $\mathbf{T}_g \in \mathbb{R}^{c \times c}$ and $\mathbf{T}_h \in \mathbb{R}^{c \times c}$, which are respectively derived from support and query grad vectors ($\mathbf{g}_* \in \mathbb{R}^{1 \times c}, * \in \{s, q\}$) and holistic target representations ($\mathbf{h}_* \in \mathbb{R}^{1 \times c}, * \in \{s, q\}$), formulated as:

$$\mathbf{T}_g = \mathrm{Softmax}(\mathbf{g}_q^\mathsf{T} \mathbf{g}_s), \quad \mathbf{T}_h = \mathrm{Softmax}(\mathbf{h}_q^\mathsf{T} \mathbf{h}_s), \tag{7}$$

where the $\mathbf{g}_*$ and $\mathbf{h}_*$ are obtained by mask average pooling ($\mathbf{MAP}$) of corresponding backbone features or gradients, formally:

$$\mathbf{g}_s = \mathbf{MAP}(\nabla_s, \mathbf{M}_s), \quad \mathbf{g}_q = \mathbf{MAP}(\nabla_q, \mathbb{M}), \tag{8}$$

$$\mathbf{h}_s = \mathbf{MAP}(\widehat{\mathbf{F}}_s, \mathbf{M}_s), \quad \mathbf{h}_q = \mathbf{MAP}(\widehat{\mathbf{F}}_q, \mathbb{M}), \tag{9}$$

among which, $\mathbf{M}_s$ is the support target mask, as in equation 2, $\mathbb{M}$ represents the predicted query target area with relatively higher confidence. The calibrated support features are obtained by:

$$\widetilde{\mathbf{F}}_s = \widehat{\mathbf{F}}_s + \widehat{\mathbf{F}}_s(\mathbf{T}_g + \mathbf{T}_h), \tag{10}$$

note that a residual connection is retained for the stable training. The transformation matrices establish channel-wise correspondences between support and query target features from both the holistic representation and the gradient perspective. Through dual transformation, CBB facilitates the task-specific category-compactness of the feature pairs, which paves the way for the processing of the query decoder. The proposed TCMNet can be easily integrated with different decoders and extended to 5-shot settings with minimal method-independent changes.

## 4 Experiments

### 4.1 Datasets and Evaluation Metrics

We evaluate the proposed TCMNet on two popular few-shot segmentation benchmarks, *i.e.*, Pascal-$5^i$ (Shaban et al., 2017) and COCO-$20^i$ (Nguyen & Todorovic, 2019). Among them, Pascal-$5^i$ is built based on the PASCAL VOC 2012 dataset (Everingham et al., 2010) with additional annotations from SBD (Hariharan et al., 2014). We follow the baseline works (Tian et al., 2020; Lang et al., 2022a; Peng et al., 2023) to divide the 20 categories into four folds, with three folds for training and one for testing. COCO-$20^i$ is a larger dataset with more categories and more complex scenes built from MSCOCO dataset (Lin et al., 2014). 80 categories are partitioned for cross-validation, with 60 classes used for training and 20 classes for testing. When testing, 1000 episodes are randomly sampled for performance evaluation. For a fair comparison, we follow the common practice to mean intersection-over-union (mIoU) and foreground-background intersection-over-union (FBIoU) as quantitative metrics.

### 4.2 Baseline Methods and Implementation Details

**Baseline Methods.** The proposed TCMNet focuses on task-specific backbone feature modulation, effectively collaborating with various query decoders to enhance FSS performance. To verify this, we conduct experiments on three different models, including **PFENet** (Tian et al., 2020), **BAM** (Lang et al., 2022a) and **HDMNet** (Peng et al., 2023). Among them, both PFENet and BAM utilize holistic support prototypes as semantic clues to guide the query decoder. Additionally, BAM introduces a base learner to explicitly alleviate the impact of overfitting on training classes. HDMNet employs the affinity learning-based decoder to fully explore pixel-level support information, which represents the most cutting-edge performance in FSS.

Table 2: Performance on Pascal-$5^i$(Shaban et al., 2017) in terms of mIoU for 1-shot and 5-shot segmentation. The best mean results are show in **bold**.

| Method | backbone | 1-shot | | | | | 5-shot | | | | |
|---|---|---|---|---|---|---|---|---|---|---|---|
| | | Fold-0 | Fold-1 | Fold-2 | Fold-3 | Mean | Fold-0 | Fold-1 | Fold-2 | Fold-3 | Mean |
| PFENet (Tian et al., 2020) | VGG16 | 56.9 | 68.2 | 54.4 | 52.4 | 58.0 | 59.0 | 69.1 | 54.8 | 52.9 | 59.0 |
| PFENet[TPAMI2020] w/ TCM | | 58.7 | 69.2 | 56.5 | 53.9 | **59.6**$_{(\uparrow 1.6)}$ | 60.9 | 71.3 | 57.1 | 54.8 | **61.0**$_{(\uparrow 2.0)}$ |
| BAM (Lang et al., 2022a) | | 63.2 | 70.8 | 66.1 | 57.5 | 64.4 | 67.4 | 73.1 | 70.6 | 64.0 | 68.8 |
| BAM[CVPR2022] w/ TCM | | 64.8 | 72.0 | 67.5 | 58.5 | **65.7**$_{(\uparrow 1.3)}$ | 69.2 | 75.1 | 72.5 | 64.8 | **70.4**$_{(\uparrow 1.6)}$ |
| HDMNet (Peng et al., 2023) | | 64.8 | 71.4 | 67.7 | 56.4 | 65.1 | 68.1 | 73.1 | 71.8 | 64.0 | 69.3 |
| HDMNet[CVPR2023] w/ TCM | | 65.6 | 72.4 | 68.4 | 60.3 | **66.6**$_{(\uparrow 1.5)}$ | 69.2 | 74.5 | 72.8 | 65.7 | **70.5**$_{(\uparrow 1.2)}$ |
| PFENet (Tian et al., 2020) | ResNet-50 | 61.7 | 69.5 | 55.4 | 56.3 | 60.8 | 63.1 | 70.7 | 55.8 | 57.9 | 61.9 |
| PFENet[TPAMI2020] w/ TCM | | 64.0 | 72.2 | 57.7 | 58.8 | **63.2**$_{(\uparrow 2.4)}$ | 65.3 | 73.1 | 58.2 | 60.5 | **64.3**$_{(\uparrow 2.4)}$ |
| BAM (Lang et al., 2022a) | | 69.0 | 73.6 | 67.6 | 61.1 | 67.8 | 70.6 | 75.1 | 70.8 | 67.2 | 70.9 |
| BAM[CVPR2022] w/ TCM | | 70.6 | 75.3 | 69.4 | 63.5 | **69.7**$_{(\uparrow 1.9)}$ | 72.7 | 77.4 | 72.6 | 69.7 | **73.1**$_{(\uparrow 2.2)}$ |
| HDMNet (Peng et al., 2023) | | 71.0 | 75.4 | 68.9 | 62.1 | 69.4 | 71.3 | 76.2 | 71.3 | 68.5 | 71.8 |
| HDMNet[CVPR2023] w/ TCM | | 72.1 | 76.8 | 71.0 | 64.7 | **71.1**$_{(\uparrow 1.7)}$ | 72.8 | 78.5 | 73.9 | 70.2 | **73.9**$_{(\uparrow 2.1)}$ |
| FPTrans (Zhang et al., 2022a) | ViT-B/16 | 67.1 | 69.8 | 65.6 | 56.4 | 64.7 | 73.5 | 75.7 | 77.4 | 68.3 | 73.7 |
| FPTrans[NeurIPS2022] w/ TCM | | 68.5 | 72.2 | 66.7 | 58.6 | **66.3**$_{(\uparrow 1.6)}$ | 75.9 | 77.5 | 79.2 | 70.2 | **75.7**$_{(\uparrow 2.0)}$ |

**Implementation Details.** For a fair comparison, the training settings of baseline methods are kept the same as original papers unless otherwise stated. We set the number of self-attention layers adopted in the self-modulation block as 3 and the embedding dimension of attention as $(hw)/4$. The $\lambda$ is set to be 0.1 to prevent excessive influence on the attention process. We calculate $\delta$ based on the average difference of foreground and background logits, specifically, $\delta = 0.1 \times \mathbf{Mean}_{(i,j)}(|\mathbf{P}_{fg}(i,j) - \mathbf{P}_{bg}(i,j)|)$. All integrated models are trained on Pascal-$5^i$ for 200 epochs and COCO-$20^i$ for 50 epochs. We use the same optimizer and learning rate as the query decoder of baseline methods. All experiments are run on four NVIDIA GeForce RTX 3090 GPUs.

## 4.3 PERFORMANCE COMPARISON AND ANALYSIS

We quantitatively compare the performance of different models with and without TCMNet across various settings. The results on Pascal-$5^i$ are in Table 2. It can be observed that TCM-Net consistently boosts the performance of all three baseline methods, which proves that TCM-Net is compatible with both prototype-based and affinity-based query decoders. For instance, when employing VGG-16 backbone, the 1-shot and 5-shot mIoU of the SOTA approach HDM-Net improve by 1.5% and 1.2%. With ResNet-50 backbone, the integration of TCMNet brings

Table 1: Performance on Pascal-$5^i$ in terms of FB-IoU for 1-shot and 5-shot segmentation.

| Method | Backbone | FB-IoU (%) | |
|---|---|---|---|
| | | 1-shot | 5-shot |
| PFENet (Tian et al., 2020) | ResNet-101 | 73.3 | 73.9 |
| PFENet[TPAMI2020] w/ TCMNet | | **74.2** | **74.4** |
| BAM (Lang et al., 2022a) | ResNet-50 | 68.2 | 70.7 |
| BAM[CVPR2022] w/ TCMNet | | **69.2** | **72.1** |
| HDMNet (Peng et al., 2023) | | 72.2 | 77.7 |
| HDMNet[CVPR2023] w/ TCMNet | | **73.1** | **79.0** |

the performance gain of 1.7% (1-shot) and 2.1% (5-shot) on HDMNet. More significant improvements on the larger backbone indicate the scalability of the TCMNet. As shown in Table 3, when tackling the larger COCO-$20^i$ dataset with more challenging scenes, TCMNet also achieves clear performance lift on all baseline models. Especially, TCMNet enhanced HDMNet surpasses the original version by 1.1% (1-shot)&1.2% (5-shot) and 1.4% (1-shot)&2.2% (5-shot) when using the VGG-16 and ResNet-50 backbones, respectively. The pronounced improvements showcase the applicability of TCMNet in complex scenarios. In addition, Table 13 shows the 1-shot and 5-shot FB-IoU increments brought by TCMNet on different baselines. It can be observed from the quantitative comparison in Figure 5 (a) that the integration of TCMNet can significantly reduce erroneous segmentation caused by cluttered backgrounds or incomplete segmentation caused by intra-class differences. We also provide comparison results with more recent works, please refer to **Appendix** for more details.

Table 3: Performance on COCO-$20^i$ (Nguyen & Todorovic, 2019) in terms of mIoU for 1-shot and 5-shot segmentation. The best mean results are show in **bold**.

| Method | Backbone | 1-shot | | | | | 5-shot | | | | |
|---|---|---|---|---|---|---|---|---|---|---|---|
| | | Fold-0 | Fold-1 | Fold-2 | Fold-3 | Mean | Fold-0 | Fold-1 | Fold-2 | Fold-3 | Mean |
| PFENet (Tian et al., 2020) | | 33.4 | 36.0 | 34.1 | 32.8 | 34.1 | 35.9 | 40.7 | 38.1 | 36.1 | 37.7 |
| PFENet[TPAMI2020] w/ TCM | | 34.8 | 37.1 | 35.3 | 34.1 | **35.3**$_{(\uparrow 1.2)}$ | 37.6 | 42.0 | 39.7 | 37.8 | **39.3**$_{(\uparrow 1.6)}$ |
| BAM (Lang et al., 2022a) | | 39.0 | 47.0 | 46.4 | 41.6 | 43.5 | 47.0 | 52.6 | 48.6 | 49.1 | 49.3 |
| BAM[CVPR2022] w/ TCM | VGG-16 | 40.1 | 48.2 | 47.9 | 43.3 | **44.9**$_{(\uparrow 1.4)}$ | 48.4 | 53.9 | 49.8 | 50.6 | **50.7**$_{(\uparrow 1.4)}$ |
| HDMNet (Peng et al., 2023) | | 40.7 | 50.6 | 48.2 | 44.0 | 45.9 | 47.0 | 56.5 | 54.1 | 51.9 | 52.4 |
| HDMNet[CVPR2023] w/ TCM | | 41.6 | 51.8 | 49.3 | 45.1 | **47.0**$_{(\uparrow 1.1)}$ | 48.1 | 57.9 | 55.5 | 53.0 | **53.6**$_{(\uparrow 1.2)}$ |
| PFENet (Tian et al., 2020) | ResNet-101 | 34.3 | 33.0 | 32.3 | 30.1 | 32.4 | 38.5 | 38.6 | 38.2 | 34.3 | 37.4 |
| PFENet[TPAMI2020] w/ TCM | | 36.5 | 35.4 | 35.1 | 31.9 | **34.7**$_{(\uparrow 2.3)}$ | 41.3 | 40.9 | 40.4 | 36.9 | **39.9**$_{(\uparrow 2.5)}$ |
| BAM (Lang et al., 2022a) | | 43.4 | 50.6 | 47.5 | 43.4 | 46.2 | 49.3 | 54.2 | 51.6 | 49.6 | 51.2 |
| BAM[CVPR2022] w/ TCM | ResNet-50 | 45.4 | 52.4 | 49.7 | 45.0 | **48.1**$_{(\uparrow 1.9)}$ | 51.9 | 56.5 | 53.8 | 51.7 | **53.5**$_{(\uparrow 2.3)}$ |
| HDMNet (Peng et al., 2023) | | 43.8 | 55.3 | 51.6 | 49.4 | 50.0 | 50.6 | 61.6 | 55.7 | 56.0 | 56.0 |
| HDMNet[CVPR2023] w/ TCM | | 45.6 | 56.1 | 52.7 | 51.4 | **51.4**$_{(\uparrow 1.4)}$ | 52.3 | 62.6 | 58.5 | 59.2 | **58.2**$_{(\uparrow 2.2)}$ |
| FPTrans (Zhang et al., 2022a) | ViT-B/16 | 39.7 | 44.1 | 44.4 | 39.7 | 42.0 | 49.9 | 56.5 | 55.4 | 53.2 | 53.8 |
| FPTrans[NeurIPS2022] w/ TCM | | 41.3 | 46.1 | 46.1 | 41.6 | **43.8**$_{(\uparrow 1.8)}$ | 41.8 | 58.8 | 57.5 | 55.3 | **55.9**$_{(\uparrow 2.1)}$ |

Table 4: Component ablations.

| SMB | CCB | mIoU | Δ |
|---|---|---|---|
| | | 60.8 | - |
| ✓ | | 62.2 | +1.4 |
| | ✓ | 61.4 | +0.6 |
| ✓ | ✓ | **63.2** | **+2.4** |

Table 5: Ablations on SMB.

| SA | G | G' | mIoU |
|---|---|---|---|
| | | | 61.4 |
| ✓ | | | 61.7 |
| ✓ | ✓ | | 62.7 |
| ✓ | ✓ | ✓ | **63.2** |

Table 6: Ablations on CCB.

| g | h | mIoU |
|---|---|---|
| | | 62.2 |
| ✓ | | 62.9 |
| | ✓ | 62.6 |
| ✓ | ✓ | **63.2** |

Table 7: Ablation of the order of the two modules.

| option | mIoU |
|---|---|
| CBB → SMB | 62.4 |
| SMB → CBB | 63.2 |

Table 8: Ablation studies on Spatially Modulation.

| Spatial-S | Spatial-Q | mIoU |
|---|---|---|
| | | 60.8 |
| ✓ | | 61.0 |
| | ✓ | 60.1 |
| ✓ | ✓ | 60.6 |

Table 9: Ablations on different modulation strategy.

| strategy | mIoU |
|---|---|
| baseline | 60.8 |
| dot | 53.8 |
| channel weight | 57.8 |
| SMB + CBB | 63.2 |

## 4.4 ABLATION STUDY

To verify the effectiveness of each component of TCMNet, we employ the PFENet as baseline to conduct a series of ablation studies on Pascal-$5^i$ using ResNet-50 backbone. In addition to component-wise ablation studies, we also conduct an in-depth analysis of the impact of the detailed design of each block.

**Component-wise Ablations.** We recall that TCMNet comprises two key components, *i.e.*, the self-modulation block (SMB) and the cross-calibration block (CCB). The corresponding ablations are presented in Table 4. Compared to the baseline, the SMB improves the performance by 1.4% mIoU, and solely inserting the CCB brings 0.6% mIoU gains as shown in the $2^{nd}$ and $3^{rd}$ rows, respectively. This demonstrates the channel-wise self-modulation of backbone features is beneficial for feature processing within the query decoder. It can be observed that combining SMB with CCB yields a performance improvement greater than the sum of their individual improvements, suggesting effective synergy between the blocks for mutual enhancement. We further analyze how the order of the two modules affects the performance as depicted in Table 7, and discover that conducting cross-calibration after feature self-modulation (SMB→CCB) leads to superior performance. This is expected as the backbone features enhanced by SMB provide more target-aware holistic representation to guide the adaptation process.

**Investigation of Self-Modulation Block.**

**Investigation of Self-Modulation Block.** In Table 5 we further investigate the impact of internal designs of the SMB. We find that vanilla channel-wise self-attention leads to performance enhancements

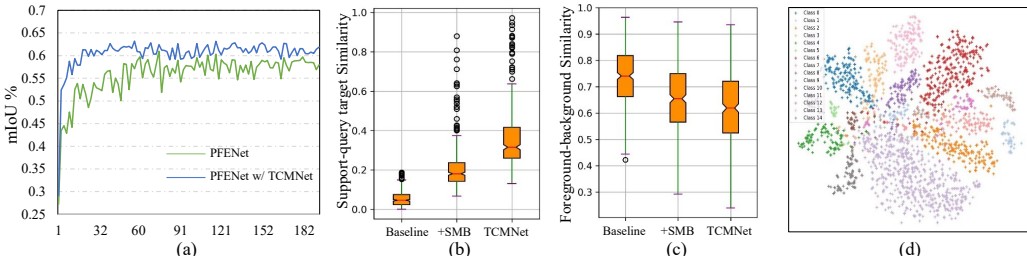

Figure 4: (a) Convergence curves of PFENet (Tian et al., 2020) with and without TCMNet. (b) Similarity distribution of support and query target target features across all test episodes, ○ denotes outlier. (c)Similarity distribution of foreground and background features across all test query images. (d) t-SNE visualization of gradient vectors of all training images.

(0.3% mIoU), indicating the benefits of capturing inter-channel dependencies. By incorporating confident target gradients $\mathbf{G}$, performance sees a notable boost of 1.0% mIoU, which we deem should be attributed to a greater focus on task-relevant meta-characteristics. As for ambiguous gradients, subtracting $\mathbf{G}'$ from the original attention matrices improves the results, suggesting that suppressing distracting channels is beneficial for target exploration. To provide a more intuitive understanding SMB, we conduct quantitative analyses to examine its impact on the discriminative capability of backbone features. As illustrated in Figure 4 (c), we analyzed the similarity distribution between background and foreground prototypes across all test samples. The SMB significantly reduces the similarity between target features and the background, thereby enhancing the target discriminability.

**Investigation of Cross-Calibration Block.** We delve into the CCB to examine the contributions of the grad vectors and the holistic representations within the dual-calibration process. As we can see from Table 6, both the gradient vectors $\mathbf{g}$ and holistic representations $\mathbf{h}$ can guide the adaptation of support features and the performance of $\mathbf{g}$ is relatively more prominent. The dual-calibration strategy formed by their combination achieves the best, which demonstrates that $\mathbf{g}$ and $\mathbf{h}$ can well bridge the intra-class target feature gap from different perspectives. We also visualize the quantitative results to analyze the impact of the CCB. As shown in Figure 4(b), after dual calibration of the support features, the similarities between the support and query target features are significantly improved. With more aligned feature pairs, the feature matching or aggregation in the query decoder achieves better correspondence, leading to improved segmentation results.

### 4.5 DISCUSSION ON TASK-SPECIFIC CHANNEL-WISE MODULATION.

We further discussed the design of TCMNet from three perspectives. **(1) Why channel-wise?** In the absence of category-aware classifiers, query decoders of current FSS models indiscriminately use all channels for feature matching. This implicit learning-based paradigm of seeking key channels not only slows down convergence, but also tends to induce channel bias toward the training categories, and the overfitting caused by channel bias is also a common issue in conventional channel attention methods, *e.g.*, SENet (Hu et al., 2018). We resort to gradients to identify the features that contribute the most to correct predictions and focus attention on them, effectively functioning as a category-aware classifier. Figure 4(a) shows that this explicit channel-wise manipulation can enable the model to converge faster to higher performance. Given the gradient guidance, we further explore the effectiveness of the self-modulation strategy in the spatial dimension. As shown in Table 8, spatial self-modulation yields a slight performance improvement when adapting to support features, while it degrades performance when applied to query features. We conjecture that this is due to gradient information focusing attention on the most discriminative regions, hindering complete target extraction. **(2) Modulation strategy.** In addition to the proposed self-modulation, we tested various methods of modulating backbone features using gradient information as shown in Table 9. We find that simply weighting features point-wise ($2^{nd}$) or channel-wise ($3^{rd}$) significantly degrade the performance. We deem the reason is that the drastic changes damage the semantic information contained in backbone features. The proposed self-modulation strategy leverages gradient information on the basis of inter-channel relationships, which approach enhances task-relevant features while preserving the original semantic information. **(3) Visualizations of gradients.** In Figure 5(b), we visualize the features of channels

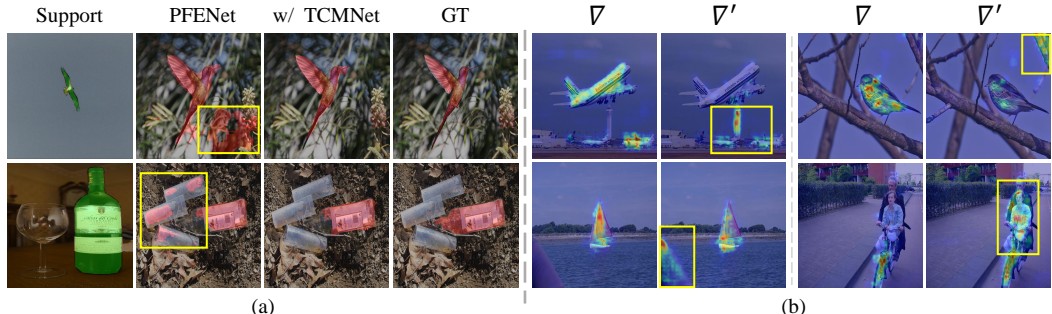

Figure 5: (a) Qualitative comparison. (b) Visualization of features with high gradients.

with high gradient values (Top 20%), It can be observed that channels with high responses in $\nabla$ and $\nabla'$ correspond to the target regions and confusing background or foreground areas, respectively, which aligns with our design intuition. We collected the gradient vectors of all category samples during training and visualized their t-SNE distributions as illustrated in Figure 4 (d). It can be observed that the gradient distributions across channels for different categories are distinctly separable, which further validates the rationale behind the motivation of TCMNet.

## 5 CONCLUSION

In this paper, we steer toward a different perspective of FSS that shifts our focus from the design of the query decoder to the better utilization of backbone features. We propose the task-specific channel-wise modulation network (TCMNet), which can serve as a generic plugin to combine with different query decoders, including a self-modulation block to enhance the target awareness of features and a cross-calibration block to bridge the intra-class variation. The decent performance on four different baseline methods indicates that exploring backbone features is another avenue for FSS in addition to query decoder design.

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

## A  APPENDIX

### A.1  COMPARISON WITH MORE RECENT METHODS.

Table 10 present the performance comparison on Pascal-5$^i$ dataset. It can be observed that the proposed TCMNet significantly outperforms previous advanced approaches and achieves new state-of-the-art results under all settings and our TCMNet can consistently boost the performance of all

Table 10: Performance comparisons with mIoU (%) as a metric on PASCAL-$5^i$, "**TCMNet** (PFENet)", "**TCMNet** (BAM)", "**TCMNet** (FPTrans)" and "**TCMNet** (HDMNet)" represent the baseline is PFENet Tian et al. (2020), BAM Lang et al. (2022a) and HDMNet Peng et al. (2023) respectively.

| Method | Backbone | 1-shot | | | | | 5-shot | | | | |
|---|---|---|---|---|---|---|---|---|---|---|---|
| | | Fold0 | Fold1 | Fold2 | Fold3 | Mean | Fold0 | Fold1 | Fold2 | Fold3 | Mean |
| SCL[CVPR2021] Zhang et al. (2021b) | Resnet-50 | 63.0 | 70.0 | 56.5 | 57.7 | 61.8 | 64.5 | 70.9 | 57.3 | 58.7 | 62.9 |
| SSP[ECCV2022] Fan et al. (2022) | Resnet-50 | 60.5 | 67.8 | 66.4 | 51.0 | 61.4 | 67.5 | 72.3 | 75.2 | 62.1 | 69.3 |
| DCAMA[ECCV2022] Shi et al. (2022b) | Resnet-50 | 67.5 | 72.3 | 59.6 | 59.0 | 64.6 | 70.5 | 73.9 | 63.7 | 65.8 | 68.5 |
| NERTNet[CVPR2022] Liu et al. (2022b) | Resnet-50 | 65.4 | 72.3 | 59.4 | 59.8 | 64.2 | 66.2 | 72.8 | 61.7 | 62.2 | 65.7 |
| IPMT[NeurIPS2022] Liu et al. (2022c) | Resnet-50 | 72.8 | 73.7 | 59.2 | 61.6 | 66.8 | 73.1 | 74.7 | 61.6 | 63.4 | 68.2 |
| ABCNet[CVPR2023] Wang et al. (2023b) | Resnet-50 | 68.8 | 73.4 | 62.3 | 59.5 | 66.0 | 71.7 | 74.2 | 65.4 | 67.0 | 69.6 |
| MIANet[CVPR2023] Yang et al. (2023) | Resnet-50 | 68.5 | 75.8 | 67.5 | 63.2 | 68.8 | 70.2 | 77.4 | 70.0 | 68.8 | 71.6 |
| MSI[ICCV2023] Moon et al. (2023) | Resnet-50 | 71.0 | 72.5 | 63.8 | 65.9 | 68.3 | 73.0 | 74.2 | 66.6 | 70.5 | 71.1 |
| AMFormer[NeurIPS2023] Wang et al. (2023a) | Resnet-50 | 71.1 | 75.9 | 69.7 | 63.7 | 70.1 | 73.2 | 77.8 | 73.2 | 68.7 | 73.2 |
| PFENet[TPAMI2023] Tian et al. (2020) | Resnet-50 | 61.7 | 69.5 | 55.4 | 56.3 | 60.8 | 63.1 | 70.7 | 55.8 | 57.9 | 61.9 |
| BAM[CVPR2022] Lang et al. (2022a) | Resnet-50 | 68.9 | 73.6 | 67.6 | 61.1 | 67.8 | 70.6 | 75.1 | 70.8 | 67.2 | 70.9 |
| FPTrans[NeurIPS2022] Zhang et al. (2022a) | ViT-B/16 | 67.1 | 69.8 | 65.6 | 56.4 | 64.7 | 73.5 | 75.7 | 77.4 | 68.3 | 73.7 |
| HDMNet[CVPR2023] Peng et al. (2023) | Resnet-50 | 71.0 | 75.4 | 68.9 | 62.1 | 69.4 | 71.3 | 76.2 | 71.3 | 68.5 | 71.8 |
| **TCMNet** (PFENet) | Resnet-50 | 64.0 | 72.2 | 57.7 | 58.8 | 63.2 | 65.3 | 73.1 | 58.2 | 60.5 | 64.3 |
| **TCMNet** (BAM) | Resnet-50 | 70.6 | 75.3 | 69.4 | 63.5 | 69.7 | 72.7 | 77.4 | 72.6 | 69.7 | 73.1 |
| **TCMNet** (FPTrans) | ViT-B/16 | 68.5 | 72.2 | 66.7 | 58.6 | 66.3 | 75.9 | 77.5 | 79.2 | 70.2 | **75.7** |
| **TCMNet** (HDMNet) | Resnet-50 | 72.1 | 76.8 | 71.0 | 64.7 | **71.1** | 72.8 | 78.5 | 73.9 | 70.2 | 73.9 |

three baseline methods with a considerable margin under all settings. Additionally, we observed that the FPTrans Zhang et al. (2022a) using ViT as the backbone performs better under the 5-shot setting. We attribute this to the global information aggregation based on attention mechanisms, which is advantageous in capturing more contextual information.

Table 11: Hyperparameter experiments on the $\lambda$.

| $\lambda$ | 0.06 | 0.08 | 0.10 | 0.12 | 0.14 |
|---|---|---|---|---|---|
| mIoU | 62.3 | 62.8 | **63.2** | 62.5 | 61.9 |

Table 12: Hyperparameter experiments on the number of layers.

| Layer | 1 | 2 | 3 | 4 | 5 |
|---|---|---|---|---|---|
| mIoU | 61.0 | 62.8 | **63.2** | 63.0 | 63.0 |

## A.2 HYPERPARAMETER EVALUATIONS.

**Evaluations of $\lambda$.** Quantitative experiments are conducted to clearly find a suitable number of $\lambda$ and the number of self-attention layers adopted in the self-modulation block. In Table 11, we report the results of different number $\lambda$ on the Pascal-$5^i$. We can find that the performance continues to grow until $\lambda = 0.10$ and then begins to decline if $\lambda$ keeps increasing. We deem the reason is that excessive interference can damage the original semantic information.

**Evaluations of the number of the self-attention layers.** As shown in Table 12, we found that when the SMB consists of only one layer, the performance is not significantly improved compared to the baseline. This is expected because, with just one layer, the gradient information does not influence the calculation

Table 13: Computational complexity and costs.

| Method | Param | GFLOPs | Time | Memory | FPS |
|---|---|---|---|---|---|
| HDMNet | 50.88M | 10.60G | 1.25 d | 6.6G | 36.4 |
| HDMNet+TCMNet | 52.12M | 12.09G | 0.5d | 7.0G | 32.0 |

of channel-wise similarity but merely alters the result of the weighted sum. However, when the number of attention layers exceeds one, there is a significant performance improvement, with the best results achieved when the number of layers is three. Therefore, we adopt three layers as the default for all experiments.

### A.3 COMPUTATIONAL COMPLEXITY AND COSTS.

Considering that our method is primarily used for modulating the backbone, the increase in the number of parameters and GFLOPs is consistent across different FSS methods. Additionally, the training time and memory usage are determined by the baseline method. Here we conducted a quantitative comparison using HDMNet as an example. We adopt four Nvidia GeForce RTX 3090 GPUs for training and one for testing.

As can be seen, our method requires only a minimal increase in the number of parameters and computational costs. Moreover, the training time can be significantly reduced as TCMNet accelerates the convergence.

