# OpenReview forum: "Task-specific Meta-feature Selection for Few-shot Segmentation"
_ICLR.cc/2025/Conference — ICLR 2025 Conference Withdrawn Submission_

### Official Review · Reviewer_hFKX · 2024-10-28

**Soundness:** 3
**Presentation:** 3
**Contribution:** 1
**Rating:** 1
**Confidence:** 5

**Summary:**

This paper presents a method for feature selection in the context of few-shot learning. The approach is built around task-specific feature selection aimed at improving model performance when limited training data is available. It focuses on leveraging specific characteristics of the support set to guide the selection process, aiming to reduce dimensionality while retaining informative features.

**Strengths:**

The paper’s presentation is good, with clear explanations of the proposed methodology, which aids in understanding the technical content.

**Weaknesses:**

The idea of feature selection in vision tasks, whether for few-shot learning or broader applications, has already been extensively explored over the years. The authors’ claim that existing methods mainly focus on designing the query decoder is misleading. In fact, there are many approaches that tackle feature selection itself in few-shot learning, as evidenced by references [1-8], which are just a few examples. By ignoring these relevant methods, the paper misses a key opportunity to clarify how it stands out. Without a deeper discussion comparing different feature selection strategies, it's impossible to see what makes this method valuable or unique.

1. Few-shot Learning for Feature Selection with Hilbert-Schmidt Independence Criterion, NeurIPS 2022.
2. Focus Your Attention when Few-Shot Classification, NeurIPS 2023
3. Class-Specific Channel Attention for Few Shot Learning, ICIP 2024
4. Boosting Few-Shot Learning via Attentive Feature Regularization, AAAI 2024
5. Cross-domain Few-shot Learning with Task-specific Adapters, CVPR 2022
6. Incremental Few-Shot Learning with Attention Attractor Networks, NeurIPS 2019
7. Meta-Learning with Attention for Improved Few-Shot Learning, ICASSAP 2021
8. Multi-attention Meta Learning for Few-shot Fine-grained Image Recognition, IJCAI 2020
9. Task-aware adaptive attention learning for few-shot semantic segmentation, Neurocomputing 2022
10. Few-shot semantic segmentation with democratic attention networks, ECCV 2020

**Questions:**

The authors need to significantly revise the entire paper, providing a comprehensive discussion of the differences between various feature-selection strategies.

---

### Official Review · Reviewer_W1RR · 2024-10-28

**Soundness:** 3
**Presentation:** 3
**Contribution:** 3
**Rating:** 5
**Confidence:** 4

**Summary:**

In this paper, for Few-shot segmentation (FSS) tasks, it is found that treating all feature channels equally is suboptimal and propose a Task-specific Channel-wise Modulation Network (TCMNet) to focus more attention on taskaware channels, facilitating more effective utilization of pre-trained features. It includes a self-modulation block to enhance the target awareness of features and a cross-calibration block to bridge the intra-class variation.

**Strengths:**

1. The motivation is clear, and the approach is sensible, straightforward, and highly applicable.
2. The paper is easy to understand, the results are easy to reproduce, and a large number of ablation experiments have been done to prove the effectiveness of the results.
3. Paper is well organized.

**Weaknesses:**

1. Is it reasonable to use confidence here to classify different regions? In fact, an easier way to think of is to use the prediction results of support to compare with the support mask to divide the regions. I don't know if the author has made relevant attempts.
2. There is a lack of theoretical explanation in the SUPPORT-TO-QUERY CROSS-CALIBRATION section. Why should this be done and why can it be done.
3. The article is similar to IFRNet [1] in terms of motivation, so it is hard to say that the motivation is novel.
[1] Yuanwei Liu, Junwei Han, Xiwen Yao, Salman Khan, Hisham Cholakkal, Rao Muhammad Anwer, Nian Liu, Fahad Shahbaz Khan:Bidirectional Reciprocative Information Communication for Few-Shot Semantic Segmentation. ICML 2024

**Questions:**

The methods compared here are all methods to freeze backbone. I would like to know whether the methods proposed by the author can be applied to the previous methods of fine-tuning backbone or adopting trainable backbone and achieve positive gains.

---

### Official Review · Reviewer_ywq6 · 2024-10-30

**Soundness:** 3
**Presentation:** 3
**Contribution:** 3
**Rating:** 5
**Confidence:** 4

**Summary:**

The paper presents a novel method to improve segmentation performance by addressing ambiguity in segmented areas. It explores the effectiveness of channel manipulation in distinguishing between foreground and background features, while acknowledging challenges like inter-class confusion. The authors suggest combining multiple tables for a comprehensive analysis and emphasize the need for more visualizations of segmentation results to enhance clarity and effectiveness.

**Strengths:**

1.	As a plug-and-play method, TCMNet achieves a solid performance improvement across various FSS methods.

**Weaknesses:**

Major

1.	The writing of the article needs to be polished. There are many unclear expressions.
2.	I think that the summary of Figure 2 is not sufficiently accurate. In fully supervised segmentation models, the fully connected layer or convolutional head serves more than just the function of channel selection.
3.	As a plug-and-play method, the additional parameter counting and FLOPs it introduces have not been thoroughly analyzed; this information should be included in the main table.
4.	From Figures 1(a) and 1(c), the effect of channel manipulation does not seem sufficiently clear, as there remains confusion among inter-class features.
5.	There are too few visualizations of the segmentation results, making it difficult to assess the effectiveness at a glance.

Minor

6.	Simplify ‘equation n’ to ‘Equ. n’.
7.	Table 1 appears after Table 2.
8.	Tables 4 to 9 can be combined into a single table with a unified caption.
9.	Part of the lines in the figures are not horizontal.

**Questions:**

1.	In section 3.2.3, could directly using the ambiguous area as a negative signal introduce noise? This is because the ambiguous area may contain both foreground and background. Is it feasible to use the background gradient as a negative signal instead?

---

### Official Review · Reviewer_FQ1F · 2024-11-04

**Soundness:** 3
**Presentation:** 3
**Contribution:** 2
**Rating:** 5
**Confidence:** 4

**Summary:**

This paper is for the few-shot segmentation (FSS) task. The proposed method focuses on better utilization of pre-trained backbone features instead of designing complex query decoders. The authors propose TCMNet, a lightweight, plug-and-play module consisting of a Self-Modulation Block that leverages gradient information to enhance task-relevant feature channels and a Cross-Calibration Block that aligns support features with query features to handle intra-class variations. The method achieves state-of-the-art performance across different backbones and decoders on both COCO-20$^i$ and Pascal-5$^i$ benchmarks while requiring less training time than the previous methods.

**Strengths:**

1.	This paper addresses an issue ignored in previous methods, i.e., the efficient utilization of pre-trained backbone features. The authors designed the TCMNet to enhance task-relevant features.
2.	The proposed TCMNet adds a low computational burden while achieving faster convergence during training; it was rarely discussed in previous FSS work.
3.	The paper is well-written and organized, easy to understand and follow.

**Weaknesses:**

1.	The authors should provide more detailed information about the training process, specifically whether the model is trained from scratch or fine-tuned on an existing one. It is important for reproducibility and understanding the proposed method.
2.	As shown in Tables 11 and 12, both the gradient information weight $\lambda$ and the number of attention layers significantly influence the final performance. Considering the fact in FSS tasks, hyperparameter selection and final performance evaluation are conducted on the same data subset without strictly separating validation and test sets, it raises concerns that the proposed method might be overfitting to the target datasets. The authors should discuss the possibility of adaptive parameter adjustment, particularly for $\lambda$, to ensure better generalization and robustness across different datasets.
3.	In Table 3, FPTrans w/ TCM shows a significant performance drop in Fold-0 under the 5-shot setting on COCO-5$^i$. The authors should analyze this and provide some possible explanations, such as whether it is related to specific classes in Fold-0.

**Questions:**

My main concern is the insufficient details regarding hyperparameter selection and training procedures. Please see the weakness part.

---

### Official Review · Reviewer_9dc4 · 2024-11-05

**Soundness:** 3
**Presentation:** 3
**Contribution:** 2
**Rating:** 3
**Confidence:** 4

**Summary:**

This work proposes Task-specific Channel-wise Modulation Network (TCMNet) to focus on task-aware channels to facilitate more effective utilization of pre-trained features for few-shot segmentation. It includes a self-modulation block to enhance the target awareness of features and a cross-calibration block to bridge the intra-class variation.

**Strengths:**

The proposed method shows consistent performance improvement on different FSS methods.

**Weaknesses:**

1. The motivation of the proposed method is unclear. This work identifies the Significant intra-class diversity between support and query targets  and Cluttered query backgrounds for few-shot segmentation. The authors propose Self-Modulation Block to solve the Cluttered query backgrounds problem, as explained ``Specifically, to deal with cluttered background, inspired by deep explanation methods (Guidotti et al., 2018), we resort to the gradient information to evaluate the importance of different channels for the current task, which is then injected into the channel-wise attention layer as explicit guidance, facilitating the concentration on the task-relevant channels.'' in L107-110. However, how does the channel-wise attention modulation can solve the cluttered background problem? The channel-wise attention cannot adjust the spatial region of the query images. Although the authors use Figure 5 (a) to show that their background prediction is improved. It is insufficient to support this argument that the channel-wise attention can solve the cluttered background problem. It only shows that TCMNet performs better on the background on this test image. Thus, it is unconvincing that the proposed method solve the claimed problems.
2. The  intra-class diversity and Cluttered query backgrounds problem has been widely explored in previous FSS methods[1,5]. However, it is unclear if the proposed method works better than other methods. It is unclear that how the proposed method solve the claimed problem. In contrast, previous FSS methods clearly show their working mechanism with sufficient analysis and experiments.
[1] Learning What Not to Segment: A New Perspective on Few-Shot Segmentation
[2] Simpler is Better: Few-shot Semantic Segmentation with Classifier Weight Transformer
[3] Few-Shot Semantic Segmentation with Cyclic Memory Network
[4] Self-Support Few-Shot Semantic Segmentation
[5] Learning Meta-class Memory for Few-Shot Semantic Segmentation
3. The authors overclaim their contribution on the exploration of fixed pre-trained backbone features. SVF [6] adjusts the backbone features in an efficient and effective manner and achieves good performance. However, the authors only mention it at L186T without any further discussion: ``Some recent works try to solve this problem by fine-tuning (Sun et al., 2022) the backbone or adopting trainable ViTs''
[6] Singular Value Fine-tuning: Few-shot Segmentation requires Few-parameters Fine-tuning
4. The novelty and contribution is limited. The proposed method is actually a channel attention method.

Overall, the proposed method works. But the authors fail to explain and investigate the working mechanism of the proposed method. It is hard to learn new insight from this work.

**Questions:**

The figure 1 is unclear. What is the baseline method? What is the meaning of different symbols?

---

### Note · Authors · 2024-11-13

I have read and agree with the venue's withdrawal policy on behalf of myself and my co-authors.